# Calciprotein Particles Cause Endothelial Dysfunction under Flow

**DOI:** 10.3390/ijms21228802

**Published:** 2020-11-20

**Authors:** Daria Shishkova, Victoria Markova, Maxim Sinitsky, Anna Tsepokina, Elena Velikanova, Leo Bogdanov, Tatiana Glushkova, Anton Kutikhin

**Affiliations:** Department of Experimental Medicine, Research Institute for Complex Issues of Cardiovascular Diseases, 6 Sosnovy Boulevard, 650002 Kemerovo, Russia; shidk@kemcardio.ru (D.S.); markve@kemcardio.ru (V.M.); sinitsky@kemcardio.ru (M.S.); cepoav@kemcardio.ru (A.T.); veliea@kemcardio.ru (E.V.); bogdla@kemcardio.ru (L.B.); glushtv@kemcardio.ru (T.G.)

**Keywords:** calciprotein particles, endothelial cells, endothelial dysfunction, shear stress, laminar flow, turbulent flow, monocyte adhesion, cell adhesion molecules, endothelial-to-mesenchymal transition, mechanotransduction

## Abstract

Calciprotein particles (CPPs), which increasingly arise in the circulation during the disorders of mineral homeostasis, represent a double-edged sword protecting the human organism from extraskeletal calcification but potentially causing endothelial dysfunction. Existing models, however, failed to demonstrate the detrimental action of CPPs on endothelial cells (ECs) under flow. Here, we applied a flow culture system, where human arterial ECs were co-incubated with CPPs for 4 h, and a normolipidemic and normotensive rat model (10 daily intravenous injections of CPPs) to simulate the scenario occurring in vivo in the absence of confounding cardiovascular risk factors. Pathogenic effects of CPPs were investigated by RT-qPCR and Western blotting profiling of the endothelial lysate. CPPs were internalised within 1 h of circulation, inducing adhesion of peripheral blood mononuclear cells to ECs. Molecular profiling revealed that CPPs stimulated the expression of pro-inflammatory cell adhesion molecules VCAM1 and ICAM1 and upregulated transcription factors of endothelial-to-mesenchymal transition (Snail, Slug and Twist1). Furthermore, exposure to CPPs reduced the production of atheroprotective transcription factors KLF2 and KLF4 and led to YAP1 hypophosphorylation, potentially disturbing the mechanisms responsible for the proper endothelial mechanotransduction. Taken together, our results suggest the ability of CPPs to initiate endothelial dysfunction at physiological flow conditions.

## 1. Introduction

Binding of ionised calcium (Ca^2+^) and phosphate by a mineral chaperone fetuin-A results in the formation of calciprotein particles (CPPs), which represent a nanoparticle vehicle mediating the clearance of excessive mineral ions from the blood [1]. Such defensive mechanism prevents medial arterial calcification (MAC) caused by hyperphosphatemia [2] and common in patients with end-stage renal disease (ESRD) [3]. MAC is aggravated in hypercalcemic conditions [4] and suppressed by magnesium supplementation [5]. If uncontrolled, MAC increases arterial stiffness [6], dramatically reduces blood vessel compliance [7], and promotes the development [8] and rupture [9] of aneurysms, being associated with higher cardiovascular mortality [10]. From the side of the media, the generation of CPPs can be considered as a protective phenomenon.

However, while guarding the organism from MAC, which develops within the years [3], CPP generation induces the development of endothelial dysfunction [11,12], a key trigger of atherosclerotic vascular disease progressing through the decades [13]. Studies in this regard indicated that CPPs cause endothelial cell (EC) death in vitro, promote the release of pro-inflammatory cytokines interleukin-6 and interleukin-8 [11,12], and cause intimal hyperplasia in both balloon-injured [11] and intact aortas [12] of rats without any background cardiovascular risk factors. A recent investigation demonstrated accelerated formation of CPPs in the blood of the individuals suffering from arterial hypertension [14]. In addition, increased CPP count was observed in patients with unstable coronary plaque phenotype and large plaque volume [15] while a higher rate of CPP generation correlated with severity and progression of coronary artery calcification [16]. Furthermore, elevated serum propensity to produce CPPs was associated with cardiovascular death and major adverse cardiovascular events in general population [17] as well as in patients with chronic kidney disease (CKD) and ESRD [18] even after the renal transplantation [19]. These studies highlighted the potential importance of CPPs for the development of cardiovascular disease; however, the molecular response of ECs to CPPs remains obscure.

In contrast to most cell types in the body, ECs perform their function under the constant blood flow that necessitates simulation of such conditions in vitro to properly interrogate endothelial physiology. Nevertheless, the majority of the studies conducted to date employed static culture for investigating the biology of ECs [20]. Accordingly, there are no reports describing pathogenic effects of CPPs on ECs cultured under flow.

Here, we utilised a commercially available flow culture system (Ibidi Pump System Quad, Ibidi, Grafelfing, Germany) and a normolipidemic rat model to explore in vitro and in vivo EC response to artificially synthesised CPPs which have been previously shown similar to those derived from the atherosclerotic plaques [11]. We focused on endothelial activation, endothelial-to-mesenchymal transition (EndoMT), and dysregulated flow sensing as the main determinants of endothelial dysfunction [13,21,22]. In contrast to the static culture, incubation of primary human coronary artery endothelial cells (HCAEC) and human internal thoracic artery endothelial cells (HITAEC) with CPPs did not lead to massive cell death. However, CPPs provoked adhesion of peripheral blood mononuclear cells (PBMCs) to the ECs, potentially through the increased expression of vascular cell adhesion molecule 1 (VCAM1) and intercellular cell adhesion molecule 1 (ICAM1). Furthermore, prolonged intravenous administration of CPPs triggered EndoMT, upregulating specific transcription factors Snail, Slug and Twist1 in vivo. Treatment with CPPs diminished the expression of mechanosensitive and atheroprotective transcription factors Krüppel-like factors 2 and 4 (KLF2 and KLF4, respectively), also reducing phosphorylation of pro-atherosclerotic transcription factor Yes-associated protein 1 (YAP1) and thereby leading to its activation which is normally suppressed at physiological flow conditions. We propose that CPPs may cause endothelial dysfunction under flow, although their deleterious effects differ across the models as well as between EC lines and arterial segments and this hypothesis clearly needs confirmation in further studies.

## 2. Results

### 2.1. Calciprotein Particles (CPPs) Are Rapidly Internalised by ECs and Promote Endothelial Dysfunction Rather than Endothelial Injury under Flow

For the adaptation of ECs to the flow, we preconditioned them by a 15 dyn/cm^2^ laminar flow for 24 h. As CPPs are believed to ripen during their circulation in vivo with the accompanying changes in their crystallinity and appearance over time, we employed both primary CPPs (CPP-P), which have a relatively amorphous structure and a spherical geometry, and secondary CPPs (CPP-S), which are crystalline and spindle- or needle-shaped (Figure 1A). As a kind of control, we also applied magnesiprotein particles (MPPs) containing magnesium, a calcium antagonist, in the complex with phosphate (Figure 1A).

Pathogenic effects of CPPs in the static culture are defined by their internalisation, which occurs as soon as they sediment onto the ECs, typically within 1 h of co-incubation [11]. However, the rate of CPP internalisation under flow remains unclear. To address this issue, we labeled CPPs with a fluorescein isothiocyanate (FITC)-labeled albumin and added them into the flow system for 1 h with a concurrent staining by LysoTracker Red, a pH sensor marking acidic organelles, and subsequent live cell imaging by means of confocal microscopy. MPP, CPP-P and CPP-S were observed in the ECs within 1 h and located in the lysosomes upon the internalisation, although co-localisation of MPPs with the lysosomes was notable only 4 h post incubation (Figure 1B).

We then asked whether CPPs inflict EC death under flow. After 4 h of co-incubation with the particles, EC monolayer was visually intact (Figure 2A), yet Hoechst 33342/ethidium bromide staining revealed a minor proportion of dead cells in cultures treated with CPP-P and CPP-S, which remained attached to the flow culture chamber (Figure 2B). As these differences were statistically significant (Figure 2C), we suggested circulating CPP-P and CPP-S as the putative triggers of EC death, although provoking endothelial dysfunction instead of endothelial injury observable under static culture conditions [11].

### 2.2. CPPs Instigate Attachment of PBMCs to the ECs

Endothelial dysfunction is generally accompanied by endothelial activation (i.e., tethering of the PBMCs by ECs), EndoMT, and impaired mechanotransduction [13,23,24]. We first assessed the ability of CPPs to trigger the adhesion of PBMCs to HCAEC and HITAEC monolayer in the flow culture chambers. Upon the addition of CellTracker Green-labeled PBMCs to the flow system for the last hour of 4-h co-incubation of ECs and CPPs, we observed a significant number of PBMCs firmly attached to the ECs (Figure 3A,B). Relative quantification of the cell adhesion molecules by Western blotting found a considerable upregulation of ICAM1 in HCAEC upon the treatment with CPP-S as well as overexpression of VCAM1 and ICAM1 in HITAEC after their incubation with CPP-P and CPP-S (Figure 3C and Appendix A). In agreement with these findings, reverse transcription quantitative polymerase chain reaction (RT-qPCR) measurements revealed increased abundance of *ICAM1* and *SELE* transcripts in CPP-treated HCAEC and upregulation of all genes responsible for the adhesion of PBMCs (*VCAM1*, *ICAM1* and *SELE*) in CPP-treated HITAEC (Figure 3D and Appendix A). Therefore, both of the EC lines were pathologically activated at the conditions of CPP circulation, although HITAEC showed an exaggerated response in comparison with HCAEC.

To better understand the relevance of the effects observed in the flow culture system for in vivo scenario free of any cardiovascular risk factors, we intravenously administered (10 daily injections) CPP-P and CPP-S to normolipidemic and normotensive Wistar rats. After 10 days, rats were euthanised with the brief flushing of the descending aorta and aortic arch (arterial segments with the laminar and turbulent flow, respectively) with TRIzol or radioimmunoprecipitation assay (RIPA) buffer containing protease and phosphatase inhibitors for the extraction of RNA and total protein, respectively. Albeit Vcam1 levels have not been found increased upon CPP-P or CPP-S treatment (Figure 3E and Appendix A), both *Vcam1* and *Icam1* genes were overexpressed in the aortic arch of CPP-S-treated rats (Figure 3F and Appendix A).

### 2.3. Circulation of CPPs May Contribute to the Development of EndoMT

We next investigated the ability of CPPs to initiate EndoMT, a process whereby ECs lose their canonical phenotype and acquire traits of mesenchymal cells, governed by the transcription factors Snail, Slug, Twist1 and Zeb1 [21,25,26]. Co-incubation of HCAEC and HITAEC with CPP-P or CPP-S in the flow culture system for 4 h did not lead to the elevation of the Snail and Slug, although a mesenchymal marker N-cadherin was higher in CPP-treated HCAEC (Figure 4A and Appendix A). Gene expression profile, however, revealed an increased level of *SNAI1* and *ZEB1* transcripts respectively in HCAEC and HITAEC exposed to the CPPs, whereas *SNAI2* transcripts were overrepresented upon the addition of CPP-P regardless of the EC line (Figure 4B and Appendix A).

Notably, endothelial lysate from the descending aorta and aortic arch of CPP-treated rats contained significantly increased levels of Snail and Slug; the latter was also characterised by an augmented Twist1 expression (Figure 4C and Appendix A). The level of α-smooth muscle actin did not increase, excluding the significant variation of vascular smooth muscle cell (VSMC)-related noise in the preparation (Figure 4C). However, gene expression profile did not point to EndoMT since genes encoding the respective transcription factor Zeb1 as well as markers of mesenchymal (Cdh2) and endothelial (Kdr) differentiation were downregulated in the descending aorta upon the treatment with CPP-P and CPP-S (Figure 4D and Appendix A). Furthermore, *Zeb1* and *Cdh2* genes were underexpressed in the aortic arch of the rats which received CPP-P injections, in contrast to *Pecam1* gene (Figure 4D and Appendix A).

### 2.4. Internalisation of CPPs Potentially Disturbs Endothelial Mechanotransduction

Endothelial mechanotransduction is mediated through a number of transcription factors including KLF2, KLF4, nuclear factor erythroid 2–related factor (NRF2), YAP1 and transcriptional co-activator with PDZ binding motif (TAZ) promoting a variety of positive and negative, although context-dependent effects [22,27,28,29]. Atheroprotective transcription factors KLF2 and KLF4 were downregulated in both EC lines, in particular upon the incubation with CPP-P (Figure 5A and Appendix A). Furthermore, phosphorylation of pro-atherogenic transcription factor YAP1 at S109 position, which ensures the sequestration of this protein in the cytosol and its transport to proteasomes, was downregulated in HCAEC after their 4-h incubation with CPPs (Figure 5A and Appendix A). Negative regulation of these molecules at the gene level was not detected (Figure 5B and Appendix A).

The results from the animal model corroborated those from in vitro experiments, demonstrating the differential downregulation of Klf2 and Klf4 in the descending aorta and aortic arch, respectively, and reduced Yap1 phosphorylation upon the CPP treatment (Figure 5C and Appendix A). Moreover, Taz was also hypophosphorylated in these conditions in the aortic arch (Figure 5C and Appendix A). Gene expression signaling was differentially regulated at CPP administration, showing downregulation of *Klf2*, *Klf4*, *Nrf2* and *Yap1* genes in the descending aorta and upregulation of *Klf2*, *Klf4* and *Wwtr1* in the aortic arch (Figure 5D and Appendix A).

## 3. Discussion

Despite the growing evidence of the CPP importance for the progression of CKD [18,19,30,31,32,33], coronary artery disease [15,16,17], and arterial hypertension [14] as well as improved understanding of their formation and maturation in the human blood, studies published to date failed to provide a significant insight into the molecular consequences of CPP internalisation. Albeit the detrimental effects of CPPs on ECs have been demonstrated previously [11,12], these studies employed a static culture model, which is canonical but does not fully correspond to the physiological conditions of EC functioning in vivo. To identify the cellular processes and pinpoint the molecular pathways involved in the response of arterial ECs to a continuous CPP circulation, here we applied a flow culture system combined with the regular systemic injections of CPPs to normolipidemic and normotensive Wistar rats. During our investigation, we focused on the pathological processes defining endothelial dysfunction (i.e., endothelial activation, EndoMT and impairment of endothelial mechanotransduction).

Distinct arterial segments have different flow patterns that significantly affect endothelial signaling, supplying ECs with opposite biomechanical cues [22,27,28,29,34]. Straight segments of the arterial tree (e.g., descending aorta) have unidirectional, laminar and atheroprotective blood flow, whereas arterial curvatures (e.g., aortic arch), bifurcations and branching points are notable for multidirectional and turbulent flow which is atheroprone [13,23,24,34]. Laminar flow ensures a proper alignment of ECs, low cell turnover, and controlled vascular permeability [13,23,24,34]. On the contrary, turbulent flow disturbs orientation and promotes regulated EC death, thereby increasing vascular permeability and contributing to the development of a pathological microenvironment [13,23,24,34]. Therefore, we collected the endothelial lysate both from descending aorta and aortic arch as the regions having laminar and turbulent flow, respectively.

Opposite to the earlier findings, CPPs did not cause massive cell death, concurrently provoking the attachment of PBMCs to the ECs in the absence of any pro-inflammatory inducers (e.g., tumor necrosis factor-α or interleukin-1β), potentially via VCAM1 and ICAM1 upregulation. In contrast to the flow culture model, lysate of rat aortic endothelium did not show elevated levels of Vcam1 protein upon the regular CPP injections, albeit being characterised by a notable overexpression of *Vcam1* and *Icam1* genes. Hence, we speculate that endothelial activation reflects initial pathological response of the endothelium to the circulation of CPPs. Augmented expression of the genes encoding pro-inflammatory cell adhesion molecules after prolonged administration of CPPs to the experimental animals suggests that endothelium remains activated in the long term, although these data need further verification by other groups and possibly utilising additional animal models.

Regular intravenous administration of CPPs induced an increase in EndoMT transcription factors in the descending aorta and particularly aortic arch; however, these effects have been limited to the gene level in the ECs cultured within the flow system. The possible explanation is that EndoMT evolves over time and enhanced transcription of the respective genes (*SNAI1*, *SNAI2* and *ZEB1*) after CPP internalisation precedes the following molecular alterations such as overexpression of Snail, Slug, and Twist1 transcription factors, which may further orchestrate the development of EndoMT.

Both ECs cultured under flow in the presence of CPPs and aortic endothelium of CPP-treated rats exhibited signs of perturbed mechanotransduction including downregulation of KLF2 and KLF4 transcription factors and loss of YAP1 suppression. These effects were particularly prominent in the aortic arch, where YAP1 was hypophosphorylated at both investigated sites (S127 and S397) and non-phosphorylated form of TAZ also prevailed indicating its resistance to degradation. Collectively, this suggests that internalisation of CPPs may deregulate endothelial mechanotransduction, instigating atheroprone phenotypic changes.

Compared to the descending aorta, endothelium of the aortic arch exhibited higher susceptibility to the systemic administration of CPPs, being notable for the elevated expression of *Vcam1* and *Icam1* genes, Twist1 upregulation and lower phosphorylation of pro-atherosclerotic transcription factors Yap and Taz. Interestingly, reduction of Klf2 levels was specific for the descending aorta while a decrease in Klf4 was characteristic of the aortic arch. Hence, flow pattern might modulate EC response to CPPs and distinct arterial segments can be differentially affected in subjects with accelerated CPP formation, e.g., those suffering from CKD. In a clinical setting, this can partially explain the subtle differences in the prevalence of cardiac and cerebrovascular events in patients with CKD [35,36,37].

In agreement with the results from the animal model, we noted a significant variation of CPP-related effects in HCAEC and HITAEC which have been isolated from atheroprone and atheroresistant arteries, respectively [38,39,40]. Compared with HITAEC, HCAEC were more sensitive to the cytotoxic effects of needle-shaped and crystalline CPP-S. At the molecular level, HCAEC responded to CPPs by the overexpression of *ICAM1*, *SELE* and *SNAI1* genes as well as hypophosphorylation of YAP1 at Ser109 residue, whereas HITAEC showed an upregulation of all genes encoding cell adhesion molecules (*VCAM1*, *ICAM1* and *SELE*) and *ZEB1* transcription factor. Therefore, EC specification, even within the same lineage, might also affect the molecular response to CPPs under flow. In vivo, this may have further implications with regard to the EC-specific paracrine effects, for instance, on underlying VSMCs. Co-culture of CPP-treated ECs with the respective VSMCs and subsequent profiling of EC and VSMC transcriptome, proteome and secretome may be considered as a promising experimental setup to test this hypothesis.

Upon the generation in the calcium- and/or phosphate-supersaturated blood, spherical and amorphous CPP-P gradually transform into the needle-shaped and crystalline CPP-S by a process termed amorphous-to-crystalline transition [41,42]. CPP-P are preferentially internalised by the ECs while CPP-S are primarily engulfed by the macrophages [43]. Differential internalisation of CPP-P and CPP-S upon *MSR1* gene knockout or blockade of the corresponding receptor indicated that binding of CPP-P and CPP-S is performed by distinct receptors whose expression seems to be cell-dependent [43]. Furthermore, CPP-P and CPP-S potentiated the release of different cytokines (interleukin-1β and tumor necrosis factor-α, respectively), conceivably due to a higher solubility of CPP-P in lysosomes and thereby more pronounced calcium-dependent inflammasome activation [43]. However, investigation of the specific effects triggered by CPP-P or CPP-S is still in its infancy. Here, we found that CPP-P upregulates *SNAI2* genes and downregulate KLF2 and KLF4 in human ECs, whereas CPP-S specifically leads to the elevated *Vcam1* and *Icam1* gene expression in the aortic arch of CPP-treated rats. Other pathological alterations were common for both CPP types, confirming a similar rate of intimal hyperplasia after CPP-P and CPP-S injections which has been previously observed in the same rat model [12]. This suggests that CPPs possess deleterious effects regardless of the amorphous-to-crystalline transition, and their formation in the blood may be detrimental for the ECs even if they are internalised in a spherical and amorphous form (CPP-P).

Although we selected a relatively short time (4 h) of EC exposure to CPPs within the flow system, internalisation of CPPs occurred within 1 h post incubation and both slight increase in dead cell count and considerable attachment of PBMCs, indicative of a pathological response, were evident in CPP-treated ECs at a 4-h time point. We therefore used a flow culture model for the simulation of the immediate EC response, whereas regular intravenous administration of CPPs (10 daily injections) was employed to test the long-term reaction of the endothelium to the increased CPP generation in vivo.

It should be noted that, at least in the Ibidi Pump System Quad setting (distinct flow culture systems might differ in this regard), alignment of ECs along the direction of flow significantly depends on their seeding density. At lower seeding density, ECs generally better elongate with the flow but often do not form a confluent monolayer whose integrity is crucial for the endothelial biology [44]. Higher seeding density (here we seeded 1/8 of confluent T-75 flask, which is equivalent to ≈ 350,000 ECs, per flow culture chamber) ensured the integrity of the endothelial monolayer yet affected the alignment of ECs, which have shown organised orientation rather than the elongation. However, we do not expect that this could affect the results, as ECs become adapted to the flow during the 24-h preconditioning.

Another limitation of our study is that the descending aorta and aortic arch, although being generally and respectively considered as the arterial segments with laminar and turbulent flow [13,23,24,34], have the intrinsic heterogeneity of the flow patterns. During early systole, aortic arch areas are exposed to laminar flow [45], while some of the regions within the descending aorta are constantly exposed to disturbed flow (e.g., branching points) [46]. Ideally, regional differences can be better studied by en face immunofluorescent staining, yet this technique is relatively low-throughput (two or three stainings per animal) and significantly complicated by the elastin autofluorescence which restricts the analysis to 2–3 Z-stacks. Imaging mass cytometry can be also suggested as an appropriate approach; however, it is expensive and not broadly available.

Further investigations regarding the EC-specific response to CPPs under flow may involve genetically engineered animal models (e.g., hyperlipidemic apoE^−/−^ or Ldlr^−/−^ mice) or prolonged incubation with CPPs in the flow system (e.g., 2 or 3 weeks) to better mimic the pathophysiological scenario in the patient with CKD. To conclude, CPPs are potentially able to provoke a pathological endothelial response under flow, which may vary among the arterial segments and cell lines but is generally not determined by the shape and crystallinity of these particles. As this is the first study reporting the effects of CPPs on ECs under flow, further investigations in this regard are necessary to confirm our findings. To better define the endothelial response to CPPs under flow and to examine which aspects of endothelial physiology are affected by CPPs, different flow culture systems may be applied and arteries from other vascular territories should be profiled.

## 4. Materials and Methods

### 4.1. Artificial Synthesis of CPPs and MPPs

To synthesise CPP-P or CPP-S, stock solutions of CaCl_2_ (21115, Sigma-Aldrich, St. Louis, MO, USA) and Na_2_HPO_4_ (94046, Sigma-Aldrich, St. Louis, MO, USA) were diluted to equal concentrations of 3 (CPP-P) or 7.5 (CPP-S) mmol/L in Dulbecco’s modified Eagle’s medium (31330038, Thermo Fisher Scientific, Waltham, MA, USA) supplemented with 10% (CPP-P) or 1% (CPP-S) fetal bovine serum (10270106, Thermo Fisher Scientific, Waltham, MA, USA). For the synthesis of MPPs, stock solutions of MgCl_2_ (97062-848, VWR, West Chester, PA, USA) and Na_2_HPO_4_ (94046, Sigma-Aldrich, St. Louis, MO, USA) were diluted to equal concentrations of 20 mmol/L in Dulbecco’s modified Eagle’s medium (31330038, Thermo Fisher Scientific, Waltham, MA, USA) supplemented with 10% fetal bovine serum (10270106, Thermo Fisher Scientific, Waltham, MA, USA). Following 24-h incubation in cell culture conditions, medium was centrifuged at 200,000× *g* for 1 h (Optima MAX-XP, 393315, Beckman Coulter, Brea, CA, USA) with the further resuspension of particles in sterile PBS (pH 7.4, 10010023, Thermo Fisher Scientific, Waltham, MA, USA) for cell culture experiments or in NaCl (Pharmsynthez, St. Petersburg, Russian Federation) for animal studies. Particle quantification was conducted utilising a microplate spectrophotometer (Multiskan Sky, 51119700DP, Thermo Scientific, Waltham, MA, USA) at a 650 nm (OD_650_) wavelength.

For the visualisation of CPPs and MPPs, we pipetted 1 µL of the particles (OD_650_ = 0.08–0.10) on a double-sided adhesive conductive carbon tape (16084-7, Ted Pella, Redding, CA, USA), dried at room temperature for 1 h, sputter coated the samples with gold and palladium (EM ACE200, Leica, Wetzlar, Germany), and carried out scanning electron microscopy (S-3400N, Hitachi, Tokyo, Japan).

### 4.2. Cell Culture and Flow Culture Model

Primary cultures of HCAEC (300K-05a, Cell Applications, San Diego, CA, USA) and HITAEC (308K-05a, Cell Applications, San Diego, CA, USA) were grown in T-75 flasks (90076, Techno Plastic Products, Trasadingen, Switzerland) according to the manufacturer’s protocols. For the experiments, HCAEC and HITAEC were seeded into the flow culture chambers (80126, Ibidi, Grafelfing, Germany) to 85–90% confluence (≈350,000 cells per flow culture chamber), cultured overnight, preconditioned by a 15 dyn/cm^2^ laminar flow (Ibidi Pump System Quad and Perfusion Set yellow/green, 10964, Ibidi, Grafelfing, Germany) for 24 h, and then exposed to 500 µL of MPPs, CPP-P, CPP-S (OD_650_ = 0.08–0.10) or PBS which were added to the cell culture medium (MesoEndo Growth Medium, 212-500, Cell Applications, San Diego, CA, USA) for 4 h. Upon the end of the incubation and thorough washing with ice-cold PBS (pH 7.4, 10010023, Thermo Fisher Scientific, Waltham, MA, USA), cells were visualised by phase contrast microscopy (20 fields of view per flow culture chamber, AxioObserver.Z1, Carl Zeiss, Oberkochen, Germany) and lysed with TRIzol Reagent (15596018, Thermo Fisher Scientific, Waltham, MA, USA) for RNA extraction or with RIPA buffer (89901, Thermo Fisher Scientific, Waltham, MA, USA) supplied with Halt protease and phosphatase inhibitor cocktail (78444, Thermo Fisher Scientific, Waltham, MA, USA) for the total protein extraction according to the manufacturer’s protocols. To collect sufficient amounts of RNA and total protein for the RT-qPCR and Western blotting analysis, we pooled the lysate from the three sequential flow culture runs into one sample. In other words, three biological replicates were pooled into one sample, but each of the flow culture experiments was performed three times nonetheless. Quantification and quality control of the isolated RNA was performed employing Qubit 4 fluorometer (Q33238, Thermo Fisher Scientific, Waltham, MA, USA), Qubit RNA BR assay kit (Q10210, Thermo Fisher Scientific, Waltham, MA, USA), Qubit RNA IQ assay kit (Q33222, Thermo Fisher Scientific, Waltham, MA, USA), Qubit RNA IQ standards for calibration (Q33235, Thermo Fisher Scientific, Waltham, MA, USA) and Qubit assay tubes (Q32856, Thermo Fisher Scientific, Waltham, MA, USA) according to the manufacturer’s protocols. Quantification of total protein was conducted using BCA Protein Assay Kit (23227, Thermo Fisher Scientific, Waltham, MA, USA) and Multiskan Sky microplate spectrophotometer (51119700DP, Thermo Fisher Scientific, Waltham, MA, USA) in accordance with the manufacturer’s protocol.

Alternatively, cells were stained with Hoechst 33342 (2 µg/mL, H3570, Thermo Fisher Scientific, Waltham, MA, USA) and ethidium bromide (10 µg/mL, IB40075, VWR, West Chester, PA, USA) for 5 min. After the thorough washing in a dye-free medium, we visualised live (Hoechst-positive) and dead (ethidium bromide-positive) cells by fluorescence microscopy (20 fields of view per flow culture chamber, AxioObserver.Z1, Carl Zeiss, Oberkochen, Germany). Quantitative image analysis was performed using the ImageJ software (National Institutes of Health, Bethesda, MD, USA).

To verify internalisation of CPPs under flow, we added a pH sensor LysoTracker Red (500 nmol/L, L7528, Thermo Fisher Scientific, Waltham, MA, USA) together with fluorescein isothiocyanate (FITC)-labeled MPPs, CPP-P, CPP-S (OD_650_ = 0.08–0.10) or PBS to the cell culture medium (MesoEndo Growth Medium, 212-500, Cell Applications, San Diego, CA, USA) for 1 or 4 h. Labeling of CPP-P and CPP-S was performed by their incubation with FITC-labeled albumin (A23015, Thermo Fisher Scientific, Waltham, MA, USA) for 1 h with the following centrifugation at 200,000× *g* for 1 h (Optima MAX-XP, 393315, Beckman Coulter, Brea, CA, USA) and three rounds of washing in a dye-free PBS to remove non-attached FITC-labeled albumin. Nuclear counterstaining was performed by the incubation with Hoechst 33342 (2 µg/mL, H3570, Thermo Fisher Scientific, Waltham, MA, USA) for 5 min. Visualisation was performed after the thorough washing with a dye-free medium using confocal microscopy (LSM 700, Carl Zeiss, Oberkochen, Germany).

For the PBMC adhesion assay, PBMCs were isolated from a healthy volunteer (A.K.) using Histopaque-1077 (10771, Sigma-Aldrich, St. Louis, MO, USA) and labeled with a CellTracker Green CMFDA Dye (5 µmol/L, C7025, Thermo Fisher Scientific, Waltham, MA, USA) during 30 min according to the respective manufacturer’s protocols. PBMCs were then added to the flow system (125,000 cells per mL of the indicated cell culture medium, 1,500,000 cells per flow system unit) for 1 h (1 h before the end of the incubation with the particles). Upon the end of the incubation and thorough washing, nuclear counterstaining was performed (Hoechst 33342, 2 µg/mL, 5 min, H3570, Thermo Fisher Scientific, Waltham, MA, USA). Combined fluorescence and phase contrast visualisation were performed after the thorough washing in a dye-free medium (20 fields of view per flow culture chamber, AxioObserver.Z1, Carl Zeiss, Oberkochen, Germany). Quantitative image analysis was performed using the ImageJ software (National Institutes of Health, Bethesda, MD, USA).

### 4.3. Animal Model

Male Wistar rats weighing 250–300 g, 12–14 weeks of age, provided by Research Institute for Complex Issue of Cardiovascular Diseases Core Facility, were used for the animal experiments (*n* = 30). Animals were allocated in the polypropylene cages (5 rats per cage) lined with wood chips and had access to the water and food (rat chow) ad libitum. Throughout the whole time of experiment, the standard conditions of the temperature (24 ± 1°C), relative humidity (55 ± 10%), and 12-h light/dark cycles were carefully maintained, and the health status of all rats was monitored daily. No randomisation was performed to allocate animals to experimental groups or cages. There were no specific inclusion or exclusion criteria. Experiments were performed in a blinded fashion. All procedures were approved by the Local Ethical Committee of the Research Institute for Complex Issues of Cardiovascular Diseases (Protocol No. 20190410, date of approval 10 April 2019) and conformed to the guidelines from Directive 2010/63/EU of the European Parliament on the protection of animals used for scientific purposes and to the National Institutes of Health Guide for the Care and Use of Laboratory Animals.

For the investigation of the endothelial molecular response to CPPs, we performed consecutive tail vein injections of CPP-P or CPP-S (900 µL of particles per injection, OD_650_ = 0.08–0.10) or equal volume of 0.9% NaCl (10 daily injections, n = 10 rats per group, 30 rats in total) without any surgical intervention. Ten days following the start of the injections, all rats were euthanised, and descending aortas and aortic arches (i.e., aortic segments with a laminar and turbulent flow, respectively) were flushed with TRIzol Reagent (15596018, Thermo Fisher Scientific, Waltham, MA, USA) for RNA extraction (5 rats per group, 15 rats in total) or with RIPA buffer (89901, Thermo Fisher Scientific, Waltham, MA, USA) supplied with Halt protease and phosphatase inhibitor cocktail (78444, Thermo Fisher Scientific, Waltham, MA, USA) for the total protein extraction (5 rats per group, 15 rats in total) according to the manufacturer’s protocols. In this model, the amount of the endothelial lysate from one descending aorta or aortic arch was sufficient to perform the RT-qPCR and Western blotting analysis; therefore, we did not pool the samples from the distinct animals. Quantification of the isolated RNA and total protein was conducted as described above.

### 4.4. RT-qPCR

Reverse transcription was carried out utilising High Capacity cDNA Reverse Transcription Kit (4368814, Thermo Fisher Scientific, Waltham, MA, USA). Gene expression was measured by RT-qPCR using the customised primers (500 nmol/L each, Evrogen, Moscow, Russian Federation, Table 1), cDNA (20 ng) and PowerUp SYBR Green Master Mix (A25778, Thermo Fisher Scientific, Waltham, MA, USA) according to the manufacturer’s protocol for Tm ≥ 60 °C (fast cycling mode). Technical replicates (*n* = 3 per each sample collected from one flow culture chamber or one aortic segment) were performed in all RT-qPCR experiments. The reaction was considered successful if its efficiency was 90–105% and R^2^ was ≥ 0.98. Quantification of the mRNA levels in the human EC lysate (*VCAM1*, *ICAM1*, *SELE*, *SNAI1*, *SNAI2*, *TWIST1*, *ZEB1*, *CDH2*, *ACTA2*, *KDR*, *PECAM1*, *CDH5*, *KLF2*, *KLF4*, *NFE2L2*, *YAP1*, *WWTR1*) and rat endothelial lysate (*Vcam1*, *Icam1*, *Twist1*, *Zeb1*, *Cdh2*, *Kdr*, *Pecam1*, *Cdh5*, *Klf2*, *Klf4*, *Nfe2l2*, *Yap1*, *Wwtr1*) was performed by using the 2^−ΔΔCt^ method. Relative transcript levels were expressed as a value relative to the housekeeping genes (*ACTB*, *GAPDH*, *B2M* for the human EC lysate; *Actb*, *Tbp*, *B2m* for the rat endothelial lysate) and to PBS (human EC lysate) or NaCl group (rat endothelial lysate) (2^−ΔΔCt^). The adjusted values were finally represented as a heat map (green, gray and red colours reflected fold change ≤ 0.50, 0.51–1.99, and ≥2.00, respectively).

### 4.5. Western Blotting

Equal amounts of protein (12 μg per sample for the human EC lysate and 15 μg per sample for the rat endothelial lysate) were mixed with NuPAGE lithium dodecyl sulfate sample buffer (NP0007, Thermo Fisher Scientific, Waltham, MA, USA) at a 4:1 ratio and NuPAGE sample reducing agent ((NP0009, Thermo Fisher Scientific, Waltham, MA, USA) at a 10:1 ratio, denatured at 99 °C for 5 min, and then loaded on a 1.5 mm NuPAGE 4–12% Bis-Tris protein gel ((NP0335BOX, Thermo Fisher Scientific, Waltham, MA, USA). The 1:1 mixture of Novex Sharp pre-stained protein standard (LC5800, Thermo Fisher Scientific, Waltham, MA, USA) and MagicMark XP Western protein standard (LC5602, Thermo Fisher Scientific, Waltham, MA, USA) was loaded as a molecular weight marker. Proteins were separated by the sodium dodecyl sulphate-polyacrylamide gel electrophoresis (SDS-PAGE) at 150 V for 2 h using NuPAGE 2-(N-morpholino)ethanesulfonic acid SDS running buffer (NP0002, Thermo Fisher Scientific, Waltham, MA, USA), NuPAGE Antioxidant (NP0005, Thermo Fisher Scientific, Waltham, MA, USA), and XCell SureLock Mini-Cell vertical mini-protein gel electrophoresis system (EI0001, Thermo Fisher Scientific, Waltham, MA, USA). Protein transfer was performed using polyvinylidene difluoride (PVDF) transfer stacks (IB24001, Invitrogen) and iBlot 2 Gel Transfer Device (Invitrogen) according to the manufacturer’s protocols using a standard transfer mode for 30–150 kDa proteins (P0—20 V for 1 min, 23 V for 4 min, and 25 V for 2 min). PVDF membranes were then incubated in iBind Flex Solution (SLF2020, Solution Kit Thermo Fisher Scientific, Waltham, MA, USA) for 1 h to prevent non-specific binding.

Blots were probed with rabbit antibodies to VCAM1 (ab134047, 1:1000, Abcam, Cambridge, UK), ICAM1 (ab109361, 1:1000, Abcam, Cambridge, UK), Snail and Slug (ab180714, 1:500, Abcam, Cambridge, UK), KLF4 (ab215036, 1:200, Abcam, Cambridge, UK), NRF2 (ab62352, 1:200, Abcam, Cambridge, UK), YAP1 (14074, 1:500, Cell Signaling Technology, Danvers, MA, USA), phospho-YAP1-Ser109 (46931, 1:500, Cell Signaling Technology, Danvers, MA, USA), phospho-YAP1-Ser127 (13008, 1:500, Cell Signaling Technology, Danvers, MA, USA), phospho-YAP1-Ser397 (13619, 1:500, Cell Signaling Technology, Danvers, MA, USA), TAZ (70148, 1:500, Cell Signaling Technology, Danvers, MA, USA), phospho-TAZ-Ser89 (59971, 1:200, Cell Signaling Technology, Danvers, MA, USA), vimentin (loading control, ab16700, 1:500, Abcam, Cambridge, UK) and VE-cadherin (loading control, 361900, 1:100, Thermo Fisher Scientific, Waltham, MA, USA), or mouse antibodies to N-cadherin (MA515633, 1:500, Thermo Fisher Scientific, Waltham, MA, USA), TWIST1 (sc-81417, 1:100, Santa Cruz Biotechnology, Dallas, TX, USA), KLF2 (NBP2-61812, 1:200, Novus Biologicals, Littleton, CO, USA), CD31 (loading control, ab9498, 1:1000, Abcam, Cambridge, UK), and GAPDH/histone H3 (loading control, ab139416, 1:250, Abcam, Cambridge, UK). Horseradish peroxidase-conjugated goat anti-rabbit (7074, Cell Signaling Technology, Danvers, MA, USA) or goat anti-mouse (AP130P, Sigma-Aldrich, St. Louis, MO, USA) secondary antibodies were used at 1:200 and 1:1000 dilution, respectively.

Incubation with the antibodies was performed using iBind Flex Solution Kit (SLF2020, Thermo Fisher Scientific, Waltham, MA, USA), iBind Flex Cards (SLF2010, Thermo Fisher Scientific, Waltham, MA, USA) and iBind Flex Western Device (SLF2000, Thermo Fisher Scientific, Waltham, MA, USA) during 3 h according to the manufacturer’s protocols. Chemiluminescent detection was performed using SuperSignal West Pico PLUS chemiluminescent substrate (34580, Thermo Fisher Scientific, Waltham, MA, USA) and C-DiGit blot scanner (LI-COR Biosciences, Linkoln, NE, USA) in a high-sensitivity mode (12-min scanning). Densitometry was performed using the ImageJ software (National Institutes of Health, Bethesda, MD, USA) using the standard algorithm (consecutive selection and plotting of the lanes with the measurement of the peak area) and subsequent adjustment to the average of three loading controls (vimentin, CD31 and VE-cadherin for the human EC lysate; Cd31, Gapdh and histone H3 for the rat endothelial lysate) and to PBS (human EC lysate) or NaCl group (rat endothelial lysate). The adjusted densitometry values were finally represented as a heat map (green, gray and red colours reflected fold change ≤ 0.75, 0.76–1.24, and ≥1.25, respectively).

### 4.6. Statistical Analysis

Statistical analysis was performed using GraphPad Prism 8 software (GraphPad Software, San Diego, CA, USA). For descriptive statistics, data were represented by the median, 25th and 75th percentiles, and range. Groups were compared by a Kruskal–Wallis test with post hoc calculation of false discovery rate (FDR) by a two-stage linear step-up procedure of Benjamini, Krieger, and Yekutieli. *Q* values (q values are the name given to the adjusted *p* values found using an optimised FDR approach) ≤ 0.05 were regarded as statistically significant. Heat maps were designed in Microsoft Excel (Microsoft Corporation, Redmond, WA, USA).

## 5. Conclusions

We propose that CPPs induce adhesion of PBMCs to the ECs, trigger endothelial-to-mesenchymal transition and impair endothelial mechanotransduction at physiological flow conditions. Collectively, this suggests CPPs as a potential cause of endothelial dysfunction in patients suffering from the disorders of mineral homeostasis.

## Figures and Tables

**Figure 1 ijms-21-08802-f001:**
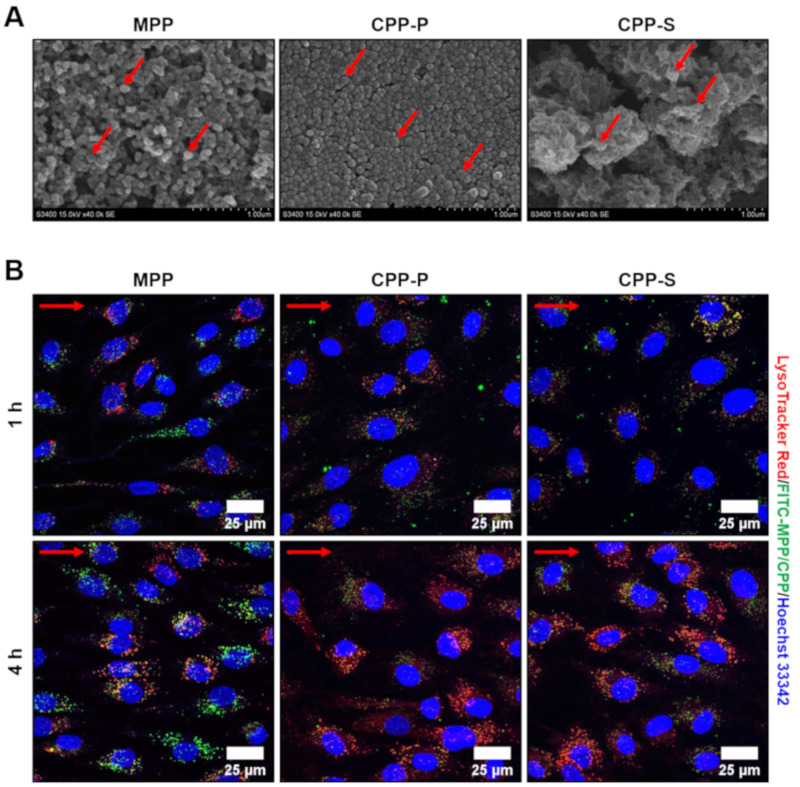
Appearance and internalisation of MPP, CPP-P and CPP-S by primary human ECs. (**A**) scanning electron microscopy of MPP, CPP-P and CPP-S, representative images, ×40,000 magnification. Red arrows indicate MPPs, CPP-P, and CPP-S; (**B**) confocal microscopy of HCAEC incubated with FITC-labeled MPP, CPP-P and CPP-S (green colour) for 1 or 4 h and additionally stained by a phago/lysosome-specific dye LysoTracker Red (red colour) along with nuclear counterstaining by Hoechst 33342 (blue colour). Note the internalisation of CPP-P and CPP-S (green dots) and their partial co-localisation with phago/lysosomes (yellow/orange dots) at both 1 and 4 h post treatment. Internalisation of MPPs (green dots) also occurred 1 h post incubation, yet co-localisation with phago/lysosomes (yellow/orange dots) was observed only at 4-h time point. Red arrows indicate the direction of flow. MPP—magnesiprotein particles, CPP-P—primary calciprotein particles, CPP-S—secondary calciprotein particles, HCAEC—human coronary artery endothelial cells, HITAEC—human internal thoracic artery endothelial cells, FITC—fluorescein isothiocyanate.

**Figure 2 ijms-21-08802-f002:**
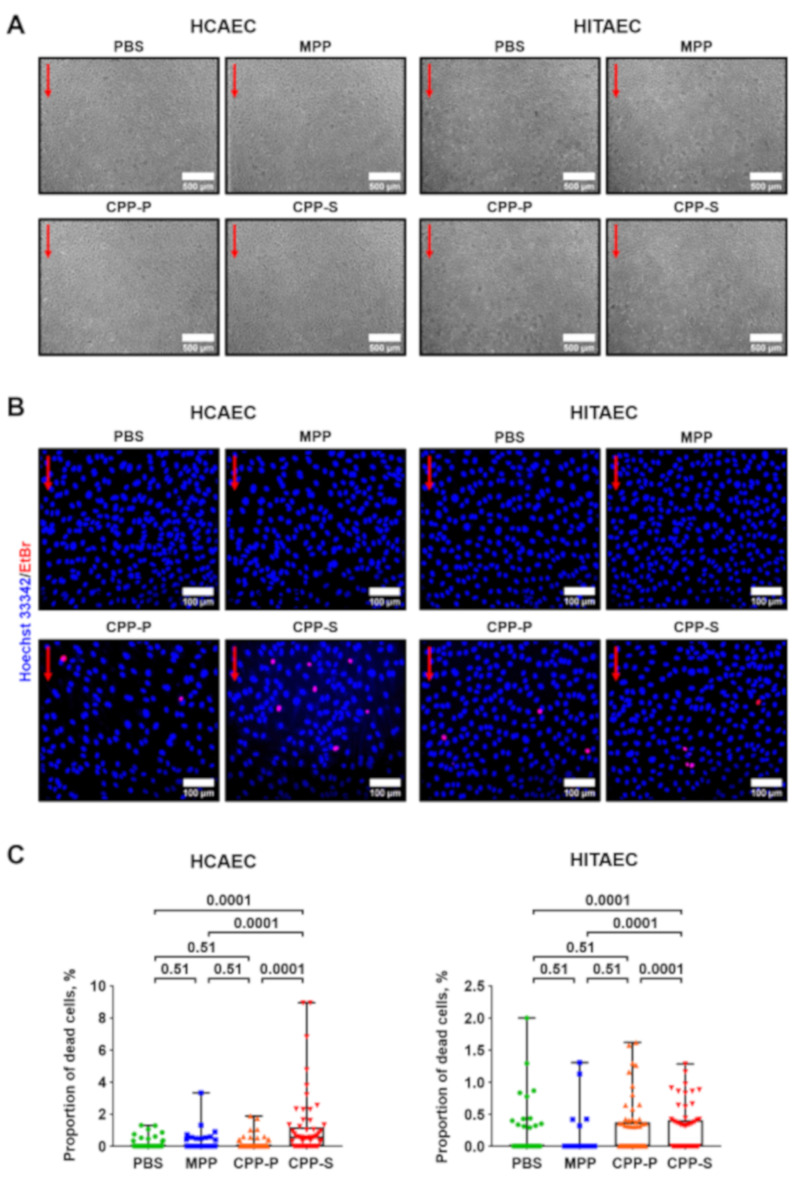
Incubation of ECs with CPPs under flow provokes endothelial dysfunction rather than injury. (**A**) phase contrast microscopy, representative images, ×50 magnification. Note the retained integrity of the EC monolayer. Red arrows indicate the direction of flow; (**B**) Hoechst 33342 (live cells, blue colour) and ethidium bromide (EtBr) staining (dead cells, red colour), representative images, ×200 magnification. Note a minor proportion of dead (EtBr-positive cells, red colour) in the cultures treated with CPP-P or CPP-S. Red arrows indicate the direction of flow; (**C**) quantification of dead cells from the experiment in (**B**), each dot represents one field of view (60 fields of view in total, 20 per flow culture chamber, three flow culture chambers per group). Whiskers indicate range, boxes bound indicate 25th–75th percentiles, center lines indicate median. *p*-values provided above boxes, Kruskal–Wallis test with post hoc false discovery rate correction by two-stage linear step-up procedure of Benjamini, Krieger and Yekutieli. HCAEC—human coronary artery endothelial cells, HITAEC—human internal thoracic artery endothelial cells, PBS—phosphate-buffered saline, MPP—magnesiprotein particles, CPP-P—primary calciprotein particles, CPP-S—secondary calciprotein particles, EtBr—ethidium bromide.

**Figure 3 ijms-21-08802-f003:**
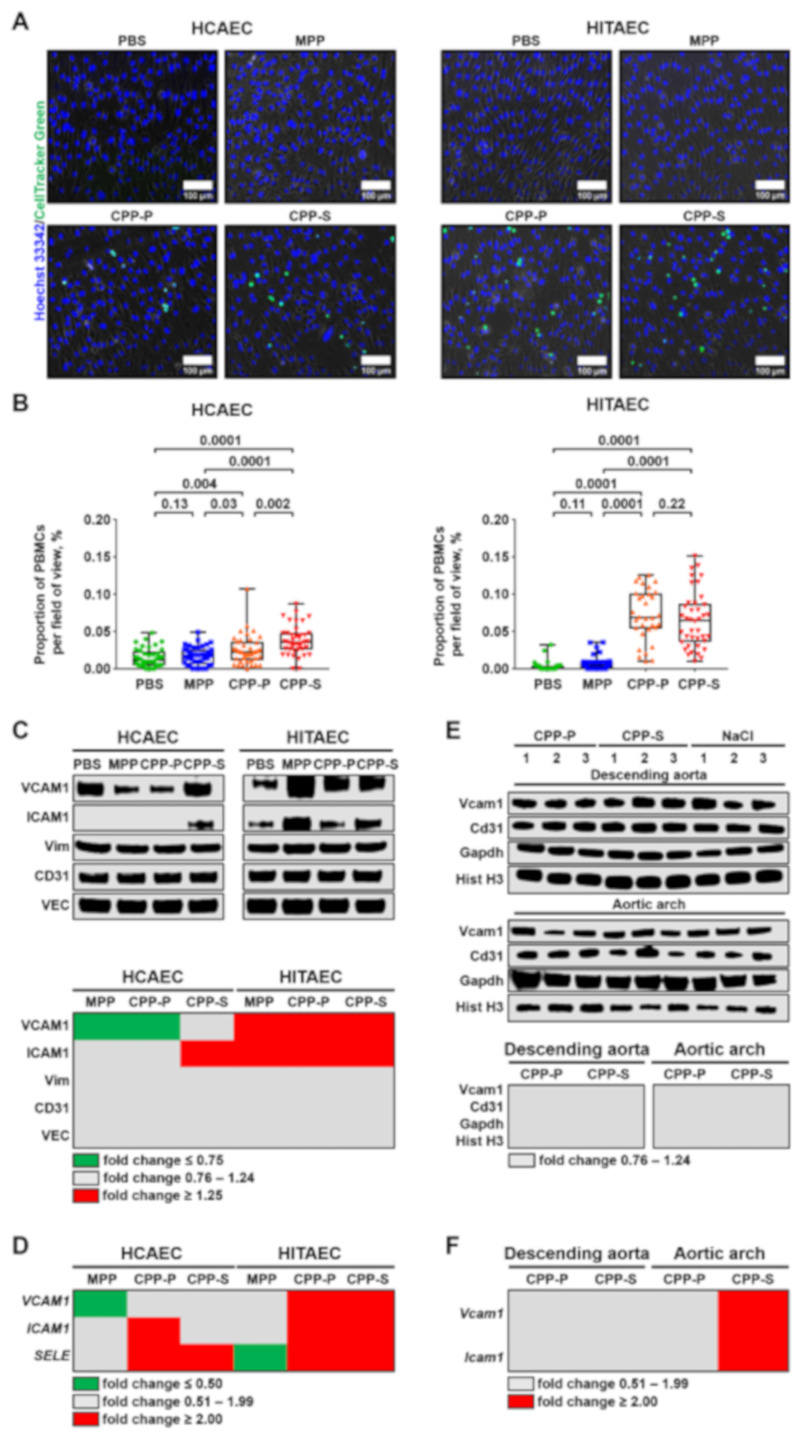
Co-incubation of ECs, PBMCs and CPPs under flow induces adhesion of PBMCs to the ECs. (**A**) HCAEC and HITAEC were exposed to CPPs circulating in the flow culture system for 4 h with the co-incubation with CellTracker Green-labeled PBMCs during the last hour. Phase contrast visualisation coupled with Hoechst 33342 (ECs and PBMCs, blue colour) and CellTracker Green (PBMCs, green colour) staining, representative images, ×200 magnification. Note the frequent adhesion of PBMCs to the CPP-P or CPP-S-treated ECs; (**B**) quantification of PBMCs attached to the ECs from the experiment in (**A**), each dot represents one field of view (40 fields of view in total, 20 per flow culture chamber, 2 flow culture chambers per group). Whiskers indicate range, boxes bounds indicate 25th–75th percentiles, center lines indicate median. *p* values provided above boxes, Kruskal–Wallis test with post hoc false discovery rate correction by two-stage linear step-up procedure of Benjamini, Krieger and Yekutieli; (**C**) Western blotting measurements of cell adhesion molecules VCAM1 and ICAM1 as compared to the expression of three EC housekeeping proteins (vimentin, CD31 and VE-cadherin) in HCAEC and HITAEC co-incubated with PBS, MPP, CPP-P or CPP-S in a flow system for 4 h. Blot scans (top) and band densitometry analysis (bottom). The results of the latter are represented by a heat map. Green, gray and red colours mean fold change ≤0.75, 0.76–1.24, and ≥1.25, respectively, compared to PBS group; (**D**) gene expression analysis of *VCAM1*, *ICAM1*, and *SELE* genes in HCAEC and HITAEC co-incubated with PBS, MPP, CPP-P or CPP-S in a flow system for 4 h. RT-qPCR measurements, the results are represented by a heat map. Green, gray and red colours mean fold change ≤ 0.50, 0.51–1.99, and ≥2.00, respectively, compared to PBS group; (**E**) Western blotting measurement of Vcam1 as compared to the expression of three EC housekeeping proteins (Cd31, Gapdh and histone H3) in the endothelial lysate collected from the descending aorta and aortic arch of Wistar rats which received consecutive tail vein injections of CPP-P, CPP-S or 0.9% NaCl (10 daily injections). Blot scans (top) and band densitometry analysis (bottom). The results of the latter are represented by a heat map. Gray colour means fold change 0.76–1.24 compared to NaCl group; (**F**) gene expression analysis of *Vcam1* and *Icam1* genes in the endothelial lysate collected from the descending aorta and aortic arch of Wistar rats which received consecutive tail vein injections of CPP-P, CPP-S or 0.9% NaCl (10 daily injections). RT-qPCR measurements, the results are represented by a heat map. Gray and red colours mean fold change 0.51–1.99 and ≥2.00, respectively, compared to NaCl group. PBMCs—peripheral blood-derived mononuclear cells, HCAEC—human coronary artery endothelial cells, HITAEC—human internal thoracic artery endothelial cells, PBS—phosphate-buffered saline, MPP—magnesiprotein particles, CPP-P—primary calciprotein particles, CPP-S—secondary calciprotein particles, VCAM1—vascular cell adhesion molecule 1, ICAM1—intercellular cell adhesion molecule 1, Vim—vimentin, CD31—cluster of differentiation 31, VE-cadherin—vascular endothelial cadherin, SELE—E-selectin, Gapdh—glyceraldehyde 3-phosphate dehydrogenase, Hist H3—histone H3, RT-qPCR—reverse transcription quantitative polymerase chain reaction.

**Figure 4 ijms-21-08802-f004:**
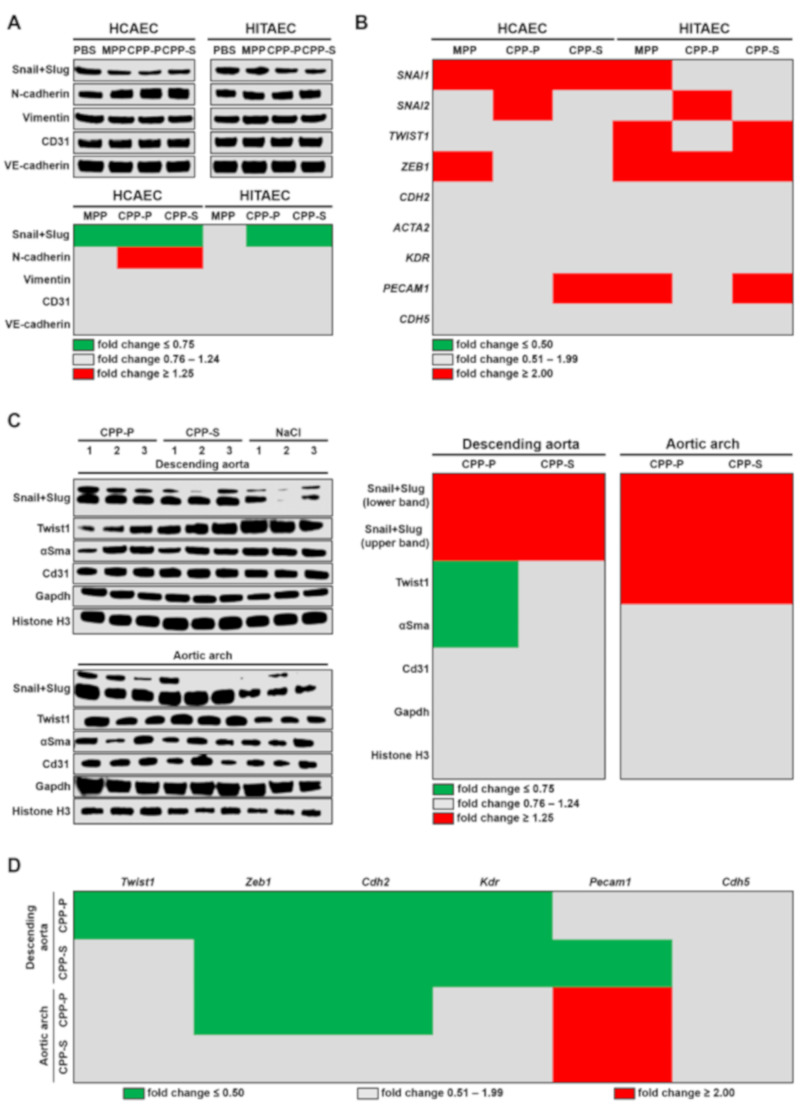
Systemic treatment with CPPs probably provokes EndoMT in rat endothelium. (**A**) Western blotting measurements of EndoMT transcription factors Snail and Slug and EndoMT marker N-cadherin as compared to the expression of vimentin, CD31 and VE-cadherin in HCAEC and HITAEC co-incubated with PBS, MPP, CPP-P or CPP-S in a flow system for 4 h. Blot scans (top) and band densitometry analysis (bottom). The results of the latter are represented by a heat map. Green, gray and red colours mean fold change ≤ 0.75, 0.76–1.24, and ≥1.25, respectively, compared to PBS group; (**B**) gene expression analysis of *SNAI1*, *SNAI2*, *TWIST1*, *ZEB1*, *CDH2*, *ACTA2*, *KDR*, *PECAM1* and *CDH5* genes in HCAEC and HITAEC co-incubated with PBS, MPP, CPP-P or CPP-S in a flow system for 4 h. RT-qPCR measurements, the results are represented by a heat map. Green, gray and red colours mean fold change ≤ 0.50, 0.51–1.99, and ≥2.00, respectively, compared to PBS group; (**C**) Western blotting measurements of EndoMT transcription factors Snail, Slug and Twist1 and α-smooth muscle actin as compared to the expression of Cd31, Gapdh and histone H3 in the endothelial lysate collected from the descending aorta and aortic arch of Wistar rats which received consecutive tail vein injections of CPP-P, CPP-S or 0.9% NaCl (10 daily injections). Blot scans (left) and band densitometry analysis (right). The results of the latter are represented by a heat map. Green, gray and red colours mean fold change ≤ 0.75, 0.76–1.24, and ≥1.25, respectively, compared to NaCl group; (**D**) gene expression analysis of *Twist1*, *Zeb1*, *Cdh2*, *Kdr*, *Pecam1*, and *Cdh5* genes in the endothelial lysate collected from the descending aorta and aortic arch of Wistar rats which received consecutive tail vein injections of CPP-P, CPP-S, or 0.9% NaCl (10 daily injections). RT-qPCR measurements, the results are represented by a heat map. Green, gray and red colours mean fold change ≤ 0.50, 0.51–1.99, and ≥2.00, respectively, compared to NaCl group. HCAEC—human coronary artery endothelial cells, HITAEC—human internal thoracic artery endothelial cells, PBS—phosphate-buffered saline, MPP—magnesiprotein particles, CPP-P—primary calciprotein particles, CPP-S—secondary calciprotein particles, N-cadherin—neural cadherin, CD31—cluster of differentiation 31, VE-cadherin—vascular endothelial cadherin, SNAI—Snail family transcriptional repressor, TWIST—Twist family basic helix-loop-helix transcription factor, ZEB—Zinc finger E-box binding homeobox, CDH—cadherin, ACTA—actin alpha, KDR—kinase insert domain receptor, PECAM—platelet endothelial cell adhesion molecule, αSma—α-smooth muscle actin, Gapdh—glyceraldehyde 3-phosphate dehydrogenase, RT-qPCR—reverse transcription quantitative polymerase chain reaction.

**Figure 5 ijms-21-08802-f005:**
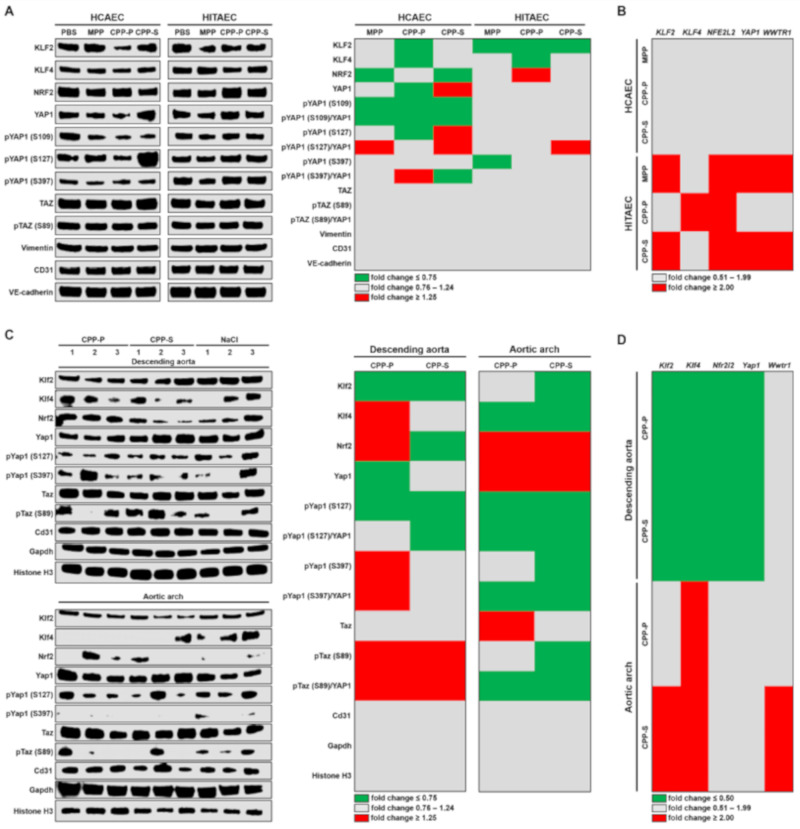
CPPs may impair endothelial mechanotransduction. (**A**) Western blotting measurements of mechanosensitive transcription factors KLF2, KLF4, NRF2, YAP1 and TAZ and phosphorylated forms of YAP1 and TAZ as compared to the expression of vimentin, CD31 and VE-cadherin in HCAEC and HITAEC co-incubated with PBS, MPP, CPP-P or CPP-S in a flow system for 4 h. Blot scans (left) and band densitometry analysis (right). The results of the latter are represented by a heat map. Green, gray and red colours mean fold change ≤ 0.75, 0.76–1.24, and ≥1.25, respectively, compared to PBS group; (**B**) gene expression analysis of *KLF2*, *KLF4*, *NFE2L2*, *YAP1*, and *WWTR1* genes in HCAEC and HITAEC co-incubated with PBS, MPP, CPP-P or CPP-S in a flow system for 4 h. RT-qPCR measurements, the results are represented by a heat map. Gray and red colours mean fold change 0.51–1.99, and ≥2.00, respectively, compared to PBS group; (**C**) Western blotting measurements of mechanosensitive transcription factors Klf2, Klf4, Nrf2, Yap1, Taz and phosphorylated forms of Yap1 and Taz as compared to the expression of Cd31, Gapdh and histone H3 in the endothelial lysate collected from the descending aorta and aortic arch of Wistar rats which received consecutive tail vein injections of CPP-P, CPP-S or 0.9% NaCl (10 daily injections). Blot scans (left) and band densitometry analysis (right). The results of the latter are represented by a heat map. Green, gray and red colours mean fold change ≤ 0.75, 0.76–1.24, and ≥1.25, respectively, compared to NaCl group; (**D**) gene expression analysis of *Klf2*, *Klf4*, *Nfe2l2*, *Yap1*, and *Wwtr1* genes in the endothelial lysate collected from the descending aorta and aortic arch of Wistar rats which received consecutive tail vein injections of CPP-P, CPP-S or 0.9% NaCl (10 daily injections). RT-qPCR measurements, the results are represented by a heat map. Green, gray and red colours mean fold change ≤ 0.50, 0.51–1.99, and ≥2.00, respectively, compared to the NaCl group. HCAEC—human coronary artery endothelial cells, HITAEC—human internal thoracic artery endothelial cells, PBS—phosphate-buffered saline, MPP—magnesiprotein particles, CPP-P—primary calciprotein particles, CPP-S—secondary calciprotein particles, KLF—Krüppel-like factor, NRF—nuclear factor erythroid 2–related factor, YAP—Yes-associated protein, pYAP—phosphorylated YAP, TAZ—transcriptional co-activator with PDZ binding motif, pTAZ—phosphorylated TAZ, CD31—cluster of differentiation 31, VE-cadherin—vascular endothelial cadherin, NFE2L2—nuclear factor erythroid 2 like 2, WWTR—WW domain containing transcription regulator, Gapdh—glyceraldehyde 3-phosphate dehydrogenase, RT-qPCR—reverse transcription quantitative polymerase chain reaction.

**Table 1 ijms-21-08802-t001:** Sequences of customised primers for RT-qPCR.

Gene	Forward Primer Sequence	Reverse Primer Sequence
**Human primers**		
*VCAM1*	5′-CGTCTTGGTCAGCCCTTCCT-3′	5′-ACATTCATATACTCCCGCATCCTTC-3′
*ICAM1*	5′-TTGGGCATAGAGACCCCGTT-3′	5′-GCACATTGCTCAGTTCATACACC-3′
*SELE*	5′-GCACAGCCTTGTCCAACC-3′	5′-ACCTCACCAAACCCTTCG-3′
*SNAI1*	5′-CAGACCCACTCAGATGTCAAGAA-3′	5′-GGGCAGGTATGGAGAGGAAGA-3′
*SNAI2*	5′-ACTCCGAAGCCAAATGACAA-3′	5′-CTCTCTCTGTGGGTGTGTGT-3′
*TWIST1*	5′-GTCCGCAGTCTTACGAGGAG-3′	5′-GCTTGAGGGTCTGAATCTTGCT-3′
*ZEB1*	5′-GATGATGAATGCGAGTCAGATGC-3′	5′-ACAGCAGTGTCTTGTTGTTGT-3′
*CDH2*	5′-GCTTCTGGTGAAATCGCATTA-3′	5′-AGTCTCTCTTCTGCCTTTGTAG-3′
*ACTA2*	5′-GTGTTGCCCCTGAAGAGCAT-3′	5′-GCTGGGACATTGAAAGTCTCA-3′
*KDR*	5′-TGCCTACCTCACCTGTTTC-3′	5′-GGCTCTTTCGCTTACTGTTC-3′
*PECAM1*	5′-TGGCGCATGCCTGTAGTA-3′	5′-TCCGTTTCCTGGGTTCAA-3′
*CDH5*	5′-AAGCGTGAGTCGCAAGAATG-3′	5′-TCTCCAGGTTTTCGCCAGTG-3′
*KLF2*	5′-CAGCACTGGTCTGGTTGCTTG-3′	5′-ACCCACTGCACACGATGCTT-3′
*KLF4*	5′-GAAAAGGACCGCCACCCACA-3′	5′-AGCGGGCGAATTTCCATCCA-3′
*NFE2L2*	5′-GCACATCCAGTCAGAAACCAGT-3′	5′-ACTGAAACGTAGCCGAAGAAAC-3′
*YAP1*	5′-AGAACTGCTTCGGCAGGCAA-3′	5′-CCACCATCCTGCTCCAGTGT-3′
*WWTR1*	5′-CGTCAGTTCCACACCAGTGC-3′	5′-GGTTCTGCTGGCTCAGGGTA-3′
*ACTB*	5′-CATCGAGCACGGCATCGTCA-3′	5′-TAGCACAGCCTGGACAGCAAC-3′
*GAPDH*	5′-AGCCACATCGCTCAGACAC-3′	5′-GCCCAATACGACCAAATCC-3′
*B2M*	5′-TCCATCCGACATTGAAGTTG-3′	5′-CGGCAGGCATACTCATCTT-3′
**Rat primers**		
*Vcam1*	5′-GGAAATGCCACCCTCACCTTA-3′	5′-TCCAGGGGAGATGTCAACAGT-3′
*Icam1*	5′-CGACATTGGGGAAGACAGCAG-3′	5′-TCCACTCGCTCTGGGAACG-3′
*Twist1*	5′-ATGTCCGCGTCCCACTAGCA-3′	5′-CCCCACGCCCTGATTCTTGT-3′
*Zeb1*	5′-CCAGTGAAGGTGATCCAGCCA-3′	5′-CTTTTTGGGTGGCGTGCAGT-3′
*Cdh2*	5′-ACCCAGGAAAAGTGGCAGGT-3′	5′-GCTGTGCTTGGCGAGTTGTC-3′
*Kdr*	5′-TGGTCCTTGCCTCAGAAGAGC-3′	5′-GCTGGTCTGGTTGGAGCCTT-3′
*Pecam1*	5′-CCGTCCAGGTGTGCGAAATG-3′	5′-GGGCGCAGTCCCATTTACTG-3′
*Cdh5*	5′-ACAAGGACGTGGTGCCAGTA-3′	5′-GGGCATCCCATTGTCGGAGA-3′
*Klf2*	5′-CACCAACTGCGGCAAGACCT-3′	5′-GTAGTGGCGGGTAAGCTCGTCA-3′
*Klf4*	5′-GACTATGCAGGCTGTGGCAAA-3′	5′-CGGTAGTGCCTGGTCAGTTCA-3′
*Nfe2l2*	5′-CAGTGGATCTGTCAGCTACTCC-3′	5′-CTCTCAACGTGGCTGGGAAT-3′
*Yap1*	5′-TTTCGGCAGGCAATACGGAAT-3′	5′-TGCTCCAGTGAGGGCAACTG-3′
*Wwtr1*	5′-GTCAACACGCCTGCCATGAA-3′	5′-TGCTCTGCTCCCGTGAATGA-3′
*Actb*	5′-ACAACCTTCTTGCAGCTCCTC-3′	5′-CCATACCCACCATCACACCCT-3′
*Tbp*	5′-TGCCAAGTGTGAGCCTCTCC-3′	5′-TGGGTTATCGTCACGCACCAT-3′
*B2m*	5′-GGTGACCGTGATCTTTCTGGTG-3′	5′-TGAGGAAGTTGGGCTTCCCATT-3′

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
