# Peer review of "Calciprotein Particles Cause Endothelial Dysfunction under Flow"

_ijms, 2020, doi:10.3390/ijms21228802_

Round 1
Reviewer 1 Report
In their MS entitled "Calciprotein Particles Cause Endothelial Dysfunction Under Current," Shishkova et al studied the effects of calciprotein particles, relevant pathological stimuli associated with mineral homeostasis disorders, on endothelial function and the induction of the inflammatory process of blood vessels. They used both an advanced in vitro endothelial cell culture model and an in vivo rat model and determined a wide range of parameters to clearly determine endothelial homeostasis disorders. Interestingly, the authors also evaluated and found the altered effects of particles on the mechanisms responsible for proper endothelial mechanotransduction. They conclude that calciprotein particles represent a significant pathological potential to initiate endothelial dysfunction under physiological flow conditions.
The article is follow up on author`s previous study “Calcium Phosphate Bions Cause Intimal Hyperplasia in Intact Aortas of Normolipidemic Rats through Endothelial Injury”. The experiments are performed well, the data is analyzed and displayed in the correct form, and the manuscript is well written. Authors should consider the following minor comments as they revise their MS.
Minor comments:
1 - There are too many references ion the general introduction part – 43 in the two small paragraph. They should be reduced significantly to the most key up to ten.
2 - Figure 1b - It would be helpful to use arrows or other markers to point out the structures that authors refer to in the text – pictures from microscope
3 - There is unclarity regarding the experiment done under flow conditions. It needs to be clarified which media/buffer was used for flow experiment to which then particles or control PBS was added.
4 - Application of particles in vivo – how were the 10 injections per day distributed in time through a day?
5 - Results row 148 – adhesion is depicted in Figure 3 b not 3c
6 - Row 208-209 the final sentence – the conclusion does not reflect results presented in figure 4d. It should be elaborated in more details.
7 - In general, the western blot results presented in figures 3, 4 and 5 do not provide and information about deviation of the data. It is necessary to make optical density determinations of western blots – at least for proteins were significant differences are suggested / concluded.
8 - Row 316 –since authors did not follow internalization in their study, it is too speculative to use it as an argument in the discussion.
9 - Statistical analysis. Since the employed method is not very typical, it would be good to state which software was used for the analysis.
Author Response
1 -There are too many references ion the general introduction part –43 in the two small paragraph. They should be reduced significantly to the most key up to ten.
1 -There are too many references ion the general introduction part –43 in the two small paragraph. They should be reduced significantly to the most key up to ten.
We agree with the reviewer that the number of references in the Introduction was excessive. We have corrected their number to 19 within the first two paragraphs and to 22within the Introduction in total. This is higher than was suggested by the reviewer (10), yet, as calciprotein particles are not broadly investigated, not many readers are familiar with the topic. Therefore, in the first part of the Introduction, we describe the importance of CPP for the human physiology and pathology as well as their clinical relevance and need tosupport every position with a reference. However, we have removed all redundant references.
2 -Figure 1b -It would be helpful to use arrows or other markers to point out the structures that authors refer to in the text –pictures from microscope.
We have indicated MPPs, CPP-P and CPP-S in the Figure 1B by the red arrows.
3 -There is unclarity regarding the experiment done under flow conditions. It needs to be clarified which media/buffer was used for flow experiment to which then particles or control PBS was added.
We have indicated the cell culture medium in the Materials and Methods (lines 440, 441, 469, 470, 480). In all flow culture experiments, we used MesoEndo Growth Medium (212-500, Cell Applications).
4 -Application of particles in vivo –how were the 10 injections per day distributed in time through a day?Actually,we performed 10 injections in total, i.e. during 10 days?
(“10 daily injections” mean one injection per day, and therefore all animals were euthanised after ten days of the experiment).
5 -Results ow 148 –adhesion is depicted in Figure 3 b not 3c. We removed that sentence as was requested by the Reviewer #2.
6 -Row 208-209 the final sentence –the conclusion does not reflect results presented in figure 4d. It should be elaborated in more details.
We agree with the reviewer and described these results more precisely (lines 211-216).
7 -In general, the western blot results presented in figures 3, 4 and 5 do not provide and information about deviation of the data. It is necessary to make optical density determinations of western blots –at leastfor proteins were significant differences are suggested / concluded.
We agree with the reviewer and provided these data (Supplementary Figures 1-3). We suggest retaining of the heat maps, as they concisely show the differences to the broad audience. Supplementary figures will provide the complete information to those readers who would wish to examine the bar graphs and statistical differences. Please note that, to collect sufficient amounts of RNA and total protein for the RT-qPCR and Western blotting analysis, we pooled the lysate from the three sequential flow culture runs into one sample. In other words, three biological replicates were pooled into one sample but each of the flow culture experiments was performed three times nonetheless. Therefore, the standard deviations are provided only for the rat model where we analysed the samples collected from the distinct animals, as the amount of the endothelial lysate from one descending aorta or aortic arch was sufficient to profile it for gene and protein expression. We have also indicated this in the Materials and Methods (lines 448-451and 511-513).
8 -Row 316 –since authors did not follow internalization in their study, it is too speculative to use it as an argument in the discussion.
We agree with the reviewer and corrected this point (lines 323-324).
9-Statistical analysis. Since the employed method is not very typical, it would be good to state which software was used for the analysis.
We agree with the reviewer and indicated the software employed for the statistical analysis and data presentation (Microsoft Excel for the heat maps and GraphPad Prism 8 for the rest of the analysis and presentation, lines 583and 588-589). We sincerely thank the reviewer for the constructive criticism and helpful comments.
Reviewer 2 Report
Shishkova et al. studied certain effects of Calciprotein particles (CPPs) on endothelial cells (ECs) cultured under laminar flow and also on rat aortic tissues. Because increased presence of CPPs in blood is associated with various human diseases including cardiovascular diseases, the study is of some interest. However, the study in its present state contains a number of weaknesses. First and foremost, various data presented are not mutually supportive. For instance, mononuclear cell adhesion onto ECs was suggested to depend on the expression of cell adhesion molecules on ECs, but the expression data on VCAM, ICAM, and SELE are not consistent with the cell adhesion data. In addition, the protein expression and gene activation data are not mutually supportive. There are other examples, which will be described below. Thus, at best, the study is preliminary and needs further work in order to establish reliable conclusions. At present, the data presented do not support the conclusion made by the authors. For this reason, the study, as it is presented, appears premature. In many experiments, it is not clear how many times they are replicated; in fact, some of them may have been done only once.
Specific comments.
- Fig. 1A. Indicate the direction of flow.
- Fig. 1. While in A, aligned cell shape by flow is obvious, cells in C, judged by the shape of nuclei, do not show any sign of alignment by flow. This is inconsistent. Figs. 2A and B do not seem to show cell alignment either.
- Lines 146-148. The authors state that ”despite MPPs also promoted the expression of VCAM1 and ICAM1 proteins in HITAEC, they neither provoked PBMC adhesion nor enhanced the expression of the respective genes (Figure 3C and 3D).” This is very intriguing and cannot be left standing like this (i.e. some answers are necessary). These results show that PBMC adhesion to ECs does not depend on the expression of VCAM and ICAM. Please explain.
- Figs. 3-5. Rather than showing heat maps, please show dot plots (or bar graphs) with statistical analysis for all these quantification data. Heat maps for these quantification are not acceptable. Please indicate the number of repeats for all these experiments.
- For analyzing ECs in situ, the arch and descending portion of aorta were separately analyzed for the protein and gene expression. One must note that each of these areas are still exposed to a mixture of different flow types. For example, certain arch areas are exposed to laminar flow, and certain areas of descending aorta are exposed to disturbed flow (such as the branch point of intercostal arteries). Regional differences can be better studied by en face immunofluorescent staining or imaging mass cytometry (IMC).
Author Response
Shishkova et al. studied certain effects of Calciprotein particles (CPPs) on endothelial cells (ECs) cultured under laminar flow and also on rat aortic tissues. Because increased presence of CPPs in blood is associated with various human diseases including cardiovascular diseases, the study is of some interest. However, the study in its present state contains a number of weaknesses. First and foremost, various data presented are not mutually supportive. Forinstance, mononuclear cell adhesion onto ECs was suggested to depend on the expression of cell adhesion molecules on ECs, but the expression data on VCAM, ICAM, and SELE are not consistent with the cell adhesion data. In addition, the protein expression and gene activation data are not mutually supportive. There are other examples, which will be described below. Thus, at best, the study is preliminary and needs further work in order to establish reliable conclusions. At present, the data presented do not support the conclusion made by the authors. For this reason, the study, as it is presented, appears premature. In many experiments, it is not clear how many times they are replicated; in fact, some of them may have been done only once.
Response to reviewersReviewer #2Shishkova et al. studied certain effects of Calciprotein particles (CPPs) on endothelial cells (ECs) cultured under laminar flow and also on rat aortic tissues. Because increased presence of CPPs in blood is associated with various human diseases including cardiovascular diseases, the study is of some interest. However, the study in its present state contains a number of weaknesses. First and foremost, various data presented are not mutually supportive. Forinstance, mononuclear cell adhesion onto ECs was suggested to depend on the expression of cell adhesion molecules on ECs, but the expression data on VCAM, ICAM, and SELE are not consistent with the cell adhesion data. In addition, the protein expression and gene activation data are not mutually supportive. There are other examples, which will be described below. Thus, at best, the study is preliminary and needs further work in order to establish reliable conclusions. At present, the data presented do not support the conclusion made by the authors. For this reason, the study, as it is presented, appears premature. In many experiments, it is not clear how many times they are replicated; in fact, some of them may have been done only once.We considerably revised the manuscript according to the reviewer’s suggestions. Here we attempted to show the pathogenic effects of CPPs for the ECs using the appropriate models simulating (flow culture system) or replicating (rat aorta) the physiological flow patterns in the human circulation. Although being physiologically relevant, these models definitely have their drawbacks which we have indicated below and also in the revised manuscript.
Specific comments.
Fig. 1A. Indicate the direction of flow.
We have indicated the direction of flow in the Figure 1A and also in Figures 1C, 2A and 2B by the red arrows.
2. Fig. 1. While in A, aligned cell shape by flow is obvious, cells in C, judged by the shape of nuclei, do not show any sign of alignment by flow. This is inconsistent. Figs. 2A and B do not seem to show cell alignment either.
We agree with the reviewer that the cell alignment along the direction of flow is obvious only in Figure 1A. Wewould indicate the following points here and in the Discussion (lines 386-394):
-At least in the Ibidi Pump System Quad setting (as we did not test all the flow culture systems, they might differ in this regard), alignment of endothelial cells along thedirection of flow significantly depends on their seeding density. At lower seeding density, ECs generally better elongate with the flow but often do not form a confluent monolayer which integrity is crucial for the endothelial biology.Higher seeding density (we seeded 1/8 of confluent T-75 flask, which is equivalent to ≈ 350,000 ECs, per flow culture chamber) ensures the integrity of the endothelial monolayer yet affecting the alignment of ECs, which show organised orientation rather than the elongation. We therefore changed the term “alignment” to “organised orientation” (Figure 1A)and clearly indicated this point in the Discussion. However, this should not affect the results, as ECs become adapted to the flow during the 24-hour preconditioning. We do have RNA-seq data comparing ECs cultured in the same flow culture chambers under static conditions and under laminar flow (24 hoursand 15 dyn/cm2as in this study), and the expression profile of the respective EC cultures is notably different. This does not relate to this studyand has not been published to date, yet we provided this material to the reviewer as an attachment(Figures R1-R4).
3. Lines 146-148. The authors state that ”despite MPPs also promoted the expression of VCAM1 and ICAM1 proteins in HITAEC, they neither provoked PBMC adhesion nor enhanced the expression of the respective genes (Figure 3C and 3D).” This is very intriguing and cannot be left standing like this (i.e. some answers are necessary). These results show that PBMC adhesion to ECs does not depend on the expression of VCAM and ICAM. Please explain.
We agree with the reviewer and removed that sentence from the manuscript, as it could confuse the readers. Actually, this could be caused by the defects in the VCAM1 and ICAM1 translocation to the EC membranebut this was not the case according to our immunofluorescence stainings of the ECs in flow culture chambers.Therefore, despite it is indeed intriguing, we have not enough arguments to explain this and should remove this sentence.
4. Figs. 3-5. Rather than showing heat maps, please show dot plots (or bargraphs) with statistical analysis for all these quantification data. Heat maps for these quantification are not acceptable. Please indicate the number of repeats for all these experiments.
We agree with the reviewer and provided these data (Supplementary Figures 1-3). We suggest retaining ofthe heat maps, asthey concisely show the differences to the broad audience. Supplementary figures will provide the complete information to those readers who would wish to examine the bar graphs and statistical differences. Please note that, to collect sufficient amounts of RNA and total protein for the RT-qPCR and Western blotting analysis, we pooled the lysate from the three sequential flow culture runsinto one sample. In other words, three biological replicates were pooled into one sample but each of the flow culture experiments was performed three times nonetheless. Therefore, the standard deviations are provided only for the rat model where we analysed the samples collected from the distinct animals, as the amount of the endothelial lysate from one descending aorta or aortic arch was sufficient to profile it for gene and protein expression.We have also indicated this in the Materials and Methods (lines 448-451 and 511-513).
5. For analyzing ECs in situ, the arch and descending portion of aorta were separately analyzed for the protein and gene expression. One must note that each of these areas are still exposed to a mixture of different flow types. For example, certain arch areas are exposed to laminar flow, and certain areas of descending aorta are exposed to disturbed flow (such as the branch point of intercostal arteries). Regional differences can be better studied by en face immunofluorescent staining or imaging mass cytometry (IMC).
We agree with the reviewer and indicated this in the Discussion (lines 395-403).We have also applied en faceimmunofluorescent staining to the rat descending aortas, yet this technique is relatively low-throughput (limited to two or three stainings per animal) and significantly complicated by the elastin autofluorescence which restricts the analysis to 2-3 Z-stacks. For the convenience of the reviewer, we provided the respective figures in the attachment(Figure R5). Imaging mass cytometry can be also suggested as an appropriate approach, however it is expensive and not broadly available.
We sincerely thank the reviewer for the constructive criticism and helpful comments.
Round 2
Reviewer 2 Report
Although the authors made modifications and added some new data, the study as a whole is still a rather weak, especially because the results are not fully supportive of conclusions of the study. AT best, this is a data paper in which effects of CPPs on ECs were studied against the background of flow. As such, the study could be considered to be novel. Data from in vitro experiments are not always consistent, but in vivo data are interesting. Due to the not so strong and mutually supporting nature of the data, the authors should done down their conclusions.
I have a few specific comments. 1) PBMCs were fluorescently labeled using FITC-conjugated albumin. Please describe how free FITC-albumin was removed. There seems to be green (FITC signal) dots that do not co-localize with lysotracker labels. Since it is known that labeled albumin does not usually go into the lysosomal compartment, the data appear to suggest that free FITC-albumin may be present in the preparation. Did the authors remove FITC-albumin that is not attached to CPPs, and if so, how? Please add this information to the method section. 2) The authors describe that ECs do not always align in the direction of flow. If this is the case in the flow culture system used, Fig. 1A is a misleading figure because this does not always seen (hence it is not a representative image). Besides, other images do not show alignment, suggesting that alignment is not a common occurrence in this flow chamber. Consider deleting this figure. In this study, alignment is not an important factor. The important thing is that ECs were exposed to flow. 3) Please tone down conclusions, and perhaps emphasize that the data was obtained from ECs exposed to flow and that certain effects of CPPs on ECs are regulated by flow.
Author Response
We sincerely thank the reviewer for the constructive criticism and helpful comments. Please see the attachment.

Round 3
Reviewer 2 Report
I have no further comments.